# Sleep Apnea and Sleep Habits: Relationships with Metabolic Syndrome

**DOI:** 10.3390/nu11112628

**Published:** 2019-11-02

**Authors:** Anne-Laure Borel

**Affiliations:** 1Department of Endocrinology, Diabetes and Nutrition, Grenoble Alpes University Hospital, 38043 Grenoble, France; alborel@chu-grenoble.fr; Tel.: +33-4-7676-55-09; 2“Hypoxia, Pathophysiology” (HP2) Laboratory INSERM U1042, Grenoble Alpes University, 38043 Grenoble, France

**Keywords:** metabolic syndrome, sleep, sleep apnea, sleep habit, sleep duration, chronotype, social jetlag

## Abstract

Excess visceral adiposity is a primary cause of metabolic syndrome and often results from excess caloric intake and a lack of physical activity. Beyond these well-known etiologic factors, however, sleep habits and sleep apnea also seem to contribute to abdominal obesity and metabolic syndrome: Evidence suggests that sleep deprivation and behaviors linked to evening chronotype and social jetlag affect eating behaviors like meal preferences and eating times. When circadian rest and activity rhythms are disrupted, hormonal and metabolic regulations also become desynchronized, and this is known to contribute to the development of metabolic syndrome. The metabolic consequences of obstructive sleep apnea syndrome (OSAS) also contribute to incident metabolic syndrome. These observations, along with the first sleep intervention studies, have demonstrated that sleep is a relevant lifestyle factor that needs to be addressed along with diet and physical activity. Personalized lifestyle interventions should be tested in subjects with metabolic syndrome, based on their specific diet and physical activity habits, but also according to their circadian preference. The present review therefore focuses (i) on the role of sleep habits in the development of metabolic syndrome, (ii) on the reciprocal relationship between sleep apnea and metabolic syndrome, and (iii) on the results of sleep intervention studies.

## 1. Introduction

Metabolic syndrome defines a group of risk factors underlying cardiovascular and metabolic diseases: abdominal obesity, atherogenic dyslipidemia, elevated blood pressure, and fasting plasma glucose [1]. The combined criteria used to define this syndrome identify a phenotype which is related to a greatest risk of developing cardiovascular and metabolic diseases than the simple addition of risks associated with each criterion [2].

Excess visceral adipose tissue is believed to be a primary driver of the cardiometabolic complications of metabolic syndrome [3]. An increase in visceral adiposity is thought to reflect the relative inability of the subcutaneous adipose tissue depot to sufficiently metabolize and store excess calories [2]. The specific characteristics of visceral adiposity, as opposed to subcutaneous adiposity elsewhere in the body, drive altered glucose homeostasis, proinflammatory adipocytokine release, and endothelial dysfunction that are the primary causes of metabolic syndrome [3].

Excess visceral fat is a modifiable risk factor that is usually associated with both excess caloric intake and a lack of physical activity [4]. Beyond these well-known, lifestyle-related etiologies, it also seems reasonable to assume that sleep habits could also contribute to abdominal obesity and, thus, metabolic syndrome. In addition, sleep apnea has been shown to increase the risk of cardiometabolic disease, and patients with metabolic syndrome are prone to develop sleep apnea [5].

The present narrative review focuses (i) on the role of sleep habits in the development of metabolic syndrome and (ii) on the reciprocal relationship between sleep apnea and metabolic syndrome. Finally, emerging evidence is reviewed regarding sleep intervention studies and how targeting of one specific lifestyle habits may impact other behaviors. 

## 2. Sleep Habits and Metabolic Syndrome

### 2.1. Definitions

Metabolic syndrome: In 1998, the World Health Organization (WHO) became the first organization to introduce the term metabolic syndrome, with a primary focus on insulin resistance and hyperglycemia [3]. In 2001, the National Cholesterol Education Program’s Adult Treatment Panel III (NCEP-ATP III) released its own definition, adding abdominal adiposity, specifically an increased waist circumference, as a major component of the syndrome [4]. Several definitions followed, issued from different societies, which mainly diverged on the clinical evaluation of abdominal adiposity. In 2009, the International Diabetes Federation Task Force on Epidemiology and Prevention; the National Heart, Lung, and Blood Institute; the American Heart Association; the World Heart Federation; and the International Atherosclerosis Society joined to release a statement harmonizing the criteria for defining the metabolic syndrome. This is the definition that is in use today, and it takes into account population-specific cutoffs for waist circumference [5]. Metabolic syndrome is defined by three of five criteria, with dyslipidemia being two criteria among: Fasting glucose ≥ 100 mg/dL or antidiabetic therapy, increased waist circumference, TG ≥ 150 mg/dl, and/or HDL-C < 40/50 in men/women or antilipidic therapy, ≥130/85 mmHg or therapy.

Sleep characteristics: Recent research has indicated that people’s sleep habits are changing. In this paper, we use the following definitions for terms commonly used in this field [6,7]:

‘Sleep duration’ is the time between falling asleep and waking up.

‘Sleep debt’ is a value calculated as the weekend sleep duration (i.e., Friday and Saturday nights) minus the sleep duration during the rest of the week.

‘Chronotype’ defines an individual’s circadian preference: ‘early’ or ‘morning’ people tend to go to bed and wake up early, whereas ‘late’ or ‘evening’ people go to bed and wake up late [8,9]. The chronotype is calculated as follows [10]: Chronotype = Mid-sleep time on free days (MSF) − 0.5 × sleep debt

MSF represents the ‘mid sleep time’ that is calculated as the mid-point between sleep onset and wake time on free days, i.e., days when people do not have to get up for work.

Whether people are a ‘night owl’ or an ‘early bird’ can also be assessed using the Morningness–Eveningness Questionnaire, a 19-item scale validated by Horne and Östberg that addresses the timing preference for different daily activities [11].

‘Social jetlag’ refers to a misalignment of sleep timing between work and free days. Work, school, and other schedules often interfere with an individual’s sleep preferences. Evening people, for example, are more likely to accumulate a sleep debt during the workdays which will be recovered during work-free days. This is calculated as the absolute difference between mid-sleep time on free days (MSF) and week days (MSW) i.e., social jetlag = |MSF − MSW| [12].

### 2.2. Epidemiological Evidence Related to Metabolic Syndrome

A normal amount of sleep is considered to be 7 to 8 h per night. Fewer than 6 h per night is classed as sleep deprivation and more than 9 h per night is considered as oversleeping. Sleep duration has decreased over the last 50 years in industrialized countries: Ιn the USA, the percentage of those who self-reported that they did not sleep enough increased during this period from about 15% to 30% [13]. In France, it is estimated that average sleep duration has shown a 1 h 30 min reduction in the last 50 years (https://www.inserm.fr/en/health-information/health-and-research-from-z/sleep). In a 2017 telephone survey to investigate public health in France called the “Baromètre de Santé publique France 2017” (Public health barometer France 2017), 12,637 men and women aged between 18 and 75 years answered questions about their sleeping habits; 36% reported sleeping fewer than 6 h per night. 

Several epidemiological studies and their meta-analysis have revealed sleep restriction to be associated with an increased prevalence of obesity [14]. Children seem particularly vulnerable: Reviews of child and adolescent data demonstrated an increased incidence of obesity in groups sleeping less than the time recommended for their age group [15]; this association was not so clearly marked in adults [16].

Regarding the specific association between sleep deprivation and metabolic syndrome, numerous studies have reported an association between a short sleep duration and an increased prevalence of metabolic syndrome. Some, but not all, research has also reported a similar association with long sleep duration. Eighteen such cross-sectional studies were included in a recent meta-analysis into (sleep) dose–(metabolic) response association [16]. A total of 75,657 adults were included, 51% of whom were men, and the age range was 18–96 years. Most studies reported odds ratios (OR) adjusted for age, sex, smoking, and alcohol intake. Short sleep was associated with metabolic syndrome, with a pooled estimate of the OR of 1.23 (95% CI, 1.11–1.37; *p* < 0.001; *I*^2^, 71%). Thus, there was a significantly higher proportion of metabolic syndrome in those who had short sleep durations, with low heterogeneity between studies. In addition, a dose–response relationship was found for durations of sleep <5 h, 5–6 h, and 6–7 h: Pooled ORs for having metabolic syndrome for these sleep groups were 1.51 (95% CI, 1.10–2.08; *p* = 0.01; *I*^2^, 88%), 1.28 (95% CI, 1.11–1.48; *p* <  0.001; *I*^2^, 67%), and 1.16 (95% CI, 1.02–1.31; *p* = 0.02; *I*^2^, 81%), respectively. There was no significant association with long sleep durations (pooled OR of 15 studies = 1.13, 95% CI, 0.97–1.32; *p* = 0.10; *I*^2^, 89%). These results were consistent with a previous meta-analysis [17].

In children, sleep deprivation has also been associated with an increase in cardiometabolic risk. This pattern has been seen in cross-sectional analyses based on both self-reported sleeping habits [18] and objective, actimetry-measured sleep patterns [19].

Although these relationships found in cross-sectional studies appear to be robust, there is less evidence for longitudinal associations. A large-scale prospective study into the risk of developing metabolic syndrome used data collected from 162,121 adults aged 20–80 years (men 47.4%) who had participated in a medical screening program in Taiwan [20]. At the start of the screening, no participant was either obese or had any characteristic of metabolic syndrome. Follow-up data were available annually from 98% of participants, and the number of visits made by each participant ranged from 2 to 19. Of these, 18.6% of people were short sleepers (<6 h/day), 72.8% were regular sleepers (6–8 h/day, control values) and 8.6% were long sleepers (>8 h/day). More than half of the participants (57.6%) reported that they had insomnia symptoms. Compared to regular sleep data, short sleep duration was associated with a 12% (adjusted HR 1.12 [1.07–1.17]) increase in risk of becoming centrally obese during the follow-up period. Short sleepers were also more likely to develop metabolic syndrome (adjusted HR 1.09 [1.05–1.13]) compared to regular sleepers (*p* < 001). By contrast, long sleep was associated with a decreased risk of metabolic syndrome (adjusted HR 0.93 [0.88–0.99]). Similar results were found in two other longitudinal studies [21,22].

It is also interesting to note that in 1344 participants of the Penn State Adult Cohort, the cardiovascular mortality associated with metabolic syndrome showed an interaction with sleep duration as measured in a single polysomnography. In this sample, the mean age was 48.8 (14.2) years, and 57.8% were women and 9.5% black. The initial prevalence of metabolic syndrome was 39.2%. Overall, those with metabolic syndrome had a higher crude mortality rate than those without (32.7% versus 15.1%); *p* < 0.01) after 16.6 (4.2) years of follow-up. The mean sleep duration for the entire sample was 5.9 (1.3) h. There was a significant interaction between metabolic syndrome and objective sleep duration for mortality risk. The risk of cardiovascular mortality associated with metabolic syndrome as a function of objective short sleep duration was 1.49 (95% CI = 0.75–2.97) for subjects who slept ≥6 h/night and 2.10 (95% CI = 1.39–3.16) for those who slept <6 h per night [23].

The impact of work on sleep quality has been shown as a major contributor on occupational health disparities. For instance, the 2011–2012 US National Health and Nutrition Examination Survey (NHANES) compared the metabolic health of 260 long-haul truck drivers from North Carolina with the general population. The results showed that more years of driving and poorer quality sleep were statistically significant predictors for the higher cardiometabolic risk that was observed in the drivers [24]. In 39,182 male employees in Japan that were followed up for up to seven years, it was also found that short sleep duration and shift work were independently associated with the development of metabolic syndrome [25].

Numerous studies have also found that subjects with late chronotypes tend to have poorer metabolic health as compared with those with early chronotypes: Τhey present more often with obesity [26,27,28], central repartition of adiposity [29,30], and with metabolic syndrome [19,30,31]. Indeed, in the Obesity, Nutrigenetics, Timing, and Mediterranean (ONTIME) study of 404 men and 1722 women, all of whom were overweight or obese, those with late chronotypes had higher BMI scores, higher triglycerides, lower HDL-cholesterol, higher levels of homeostasis model assessment for insulin resistance (HOMA-IR), and higher total metabolic syndrome scores [31].

In 1620 subjects derived from the Ansan cohort of the Korean Genome Epidemiology Study(KoGES), metabolic characteristics, body composition assessed by Dual-energy X-ray absorptiometry (DEXA), and visceral adiposity measured by computed tomography were collected and compared according to “morningness/eveningness” preference (Horne-Ostberg Morningness-Eveningness Questionnaire). In men, eveningness was associated with higher triglycerides levels but surprisingly lower systolic blood pressure. Anthropometrics measurements found less muscle mass in men with evening preference. In women, eveningness was associated with lower HDL-cholesterol and higher triglycerides and C- reactive protein (CRP) levels. Anthropometrics showed more visceral fat in women with evening preference [30].

Social jetlag, which is of greater amplitude in subjects with late chronotype, has also been independently linked with obesity [12,29] and metabolic disturbances [7]. However, few studies did not find such an association [32,33]. For instance, chronotype and social jetlag, objectively measured by wrist actimetry in 390 healthy young adults (21–35 years old), did not show any association with excess body weight, nor with elevated blood pressure [32].

### 2.3. Mechanisms of Action

Environmental cues, or “natural zeitgebers”, like the alternation between day and night or cycles in external temperatures, usually synchronize circadian rhythms including patterns of rest and activity, food intake, and daily variations in metabolic fluxes and levels of hormones [34]. The timing of such patterns also varies with a normal distribution according to individuals’ circadian preferences or chronotype. This normal distribution is in part dependent on genetic variations. In a twin study that measured circadian variations by wrist temperature, the results showed that between 46% and 70% of the observed circadian variance in temperatures could be attributed to genetic factors [35]. Three genome wide association studies (GWAS) performed in participants of European descent using data from the UK Biobank and the US genetics company 23andMe have also reported several single-nucleotide polymorphisms (SNPs) that are associated with chronotype [36,37,38]. Variations in other sleep characteristics, including sleep duration, have also been linked with polymorphisms at other loci [39].

However, sleep timing is also dependent upon extrinsic or “social zeitgebers” which may be related to work or leisure activities whose schedules conflict with intrinsic factors. A typical example of this conflict is found in shift workers: The work activities, food intake, and exposure to artificial lights at night cause a loss of internal synchrony and, in consequence, adverse effects on body weight and metabolism [40].

It has been shown that subjects with late chronotype have a tendency to eat less in the morning, while in the evening, they have higher global energy intakes and also higher intakes in sucrose, fat, and saturated fatty acids than subjects with morning chronotypes [41]. A daily caloric distribution with larger evening meals has been associated with a higher risk of obesity, particularly in those with evening chronotypes [42]. In patients with type 2 diabetes, late chronotypes have been associated with higher levels of glycated hemoglobin compared to early chronotypes and also with behaviors such as more frequently skipping breakfast [43]. Furthermore, food addiction, defined as an addictive behavior towards palatable foods, seems to be more frequent in subjects with late chronotype and is reported to be mediated by a higher frequency of insomnia and impulsivity [44]. Multiple studies have shown that an evening chronotype is associated with unhealthier diets and behaviors, such as smoking or drinking more alcohol [45,46,47]. People who are evening chronotypes are also more likely to be less physically active and to have more sedentary activities [45,47,48,49].

The role of gene variants that are associated with a latter chronotype to explain higher BMI and unfavorable metabolic traits has been evaluated in the GWAS based on UK-biobank data [37]. A reciprocal Mendelian randomization analysis, using a genetic risk score based on the 13 known variants of chronotype-linked genes, found no consistent evidence that early or late chronotypes led to higher BMI. This contrasts with another study that showed that a genetic risk score (GRS) for longer sleep duration was negatively associated with obesity [50]. 

The ONTIME study, reported by Vera et al. [31], addressed the question of the respective role gene versus behaviors to explain the deleterious metabolic profile of subjects with late chonotype. Firstly, the study showed that the GRS related to late chronotype, derived from GWAS studies, provided a reliable indication of subjects’ circadian preferences. Second, the study identified that people with late chronotype had a higher metabolic risk score than people with an early chronotype. The analyses then showed that lifestyle factors, not chronotype GRS, underlay the relationship between evening chronotypes and metabolic alterations. Late chronotypes ate all three main meals later in the day, were more likely to have larger portion sizes, second helpings, to choose energy-dense foods, and to have a higher emotional eating score. However, evening types did not have a higher caloric intake, but they were less physically active and spent longer sitting down each day. These data therefore suggest that while the GRS can capture late chronotype, it does not associate with their metabolic risks. Metabolic alterations in people with late chronotype seem linked to unhealthy behaviors rather than to genetic predisposition.

Therefore, based on current knowledge, it seems that extrinsic factors linked to circadian preference, rather than intrinsic factors, are implicated in the cardiometabolic risk profile of an individual (Figure 1).

However, there are several data that suggest a gene–environment interaction between genetic traits that underlie our sleep characteristics and the risk of developing metabolic diseases under exposition to unhealthy food or physical inactivity. For instance, in the PREVIMED study, a significant association between the CLOCK-rs4580704 SNP and the risk of developing type 2 diabetes was observed after 4.8 years of follow-up. A gene–diet effect was found since only patients in the intervention group had a protective effect of the G variant for type 2 diabetes incidence and not the control group [51].

Dashti et al. [52] performed cross-sectional meta-analyses of population-based cohorts using data from the CHARGE (Cohorts for Heart and Aging Research in Genomic Epidemiology) Consortium. They studied whether dietary intake and sleep duration modified associations between five common circadian-related gene variants (CLOCK-rs1801260, CRY2-rs11605924, MTNR1B-rs1387153, MTNR1Brs10830963, and NR1D1-rs2314339) and glycemic traits, anthropometrics, and HDL-c levels. They found that higher intakes of carbohydrates and lower intakes of fat were linked to lower fasting glucose and HOMA-IR levels. Both short and long sleep durations were associated with higher fasting glucose levels, increased BMI, and greater waist circumference. Accordingly, known associations of selected SNPs on cardiometabolic traits were essentially replicated, but no diet–gene or sleep–gene interactions were found at the prespecified Bonferroni-corrected significance level of *p* < 0.003.

Thus, some evidence argues for a gene–environment interaction with circadian-related genes. It suggests a different metabolic answer to unhealthy behaviors as well as a different benefit from lifestyle interventions according to circadian genetic traits. If confirmed in further studies, such interactions will allow personalized risk prediction and personalized lifestyle intervention.

## 3. Obstructive Sleep Apnea and Metabolic Syndrome

### 3.1. Definition

Obstructive sleep apnea syndrome (OSAS) is caused by the complete or partial collapse of the pharynx repeatedly during sleep. These repeated collapses have four main consequences: desaturation–reoxygenation sequences, transitory episodes of hypercapnia, increased respiratory effort, and repeated micro-awakenings that end the respiratory event. Central obesity predisposes individuals to OSAS due to an infiltration of fat in the neck that causes upper airway collapse and increases abdominal pressure, leading to a reduction in lung volume. It has also been suggested that adipose tissue accumulation alters the neuromechanical control of the upper airway via the effects of leptin on central respiratory drive [5,53].

OSAS is defined as the presence of apnea (a 10-second interruption in airflow) and/or hypopnea (a ≤30% decrease in respiratory airflow with an associated oxygen desaturation >3% and/or micro-arousals). It is considered as mild if the Apnea–Hypopnea Index (AHI: the sum of the number of apnea and hypopnea events per hour) is between 5 and 14.9, as moderate if the score is between 15 and 29.9, and severe if the score is >30 events per hour. Clinical signs of OSAS include daytime fatigue and sleepiness, severe and daily snoring, reported sensations of choking or suffocation during sleep, nycturia, and morning headaches. The associated daytime loss of vigilance can be dangerous due to the risk of falling asleep while driving or at work [54,55]. It also causes cognitive disorders as a result of loss of attention, memory, and concentration [56].

### 3.2. Epidemiological Evidence Relating OSAS to Metabolic Syndrome

Metabolic syndrome and OSAS share a common risk factor, abdominal obesity, and 50%–60% of people with metabolic syndrome also have OSAS [57,58]. However, studies have also shown an association, independent of obesity, between OSAS and cardiometabolic risk factors, including hypertension [59], insulin resistance [60], and type 2 diabetes [61]. OSAS has been found to be associated with metabolic syndrome in several case-control and cross-sectional studies: A meta-analysis of 13 studies (*n* = 7934 subjects) reported an increased risk of metabolic syndrome in patients with OSAS with a pooled odds ratio (OR) of 1.72 (95% CI: 1.31–2.26, *p* < 0.001) and with a BMI-adjusted pooled OR of 1.97 (95% CI: 1.34–2.88, *p* < 0.001) [62].

A recent cohort study evaluated the influence of OSAS on the incidence of metabolic syndrome in a multiethnic sample of 1853 people from two population-based samples (Episono in Brazil and HypnoLaus in Switzerland) [63]. Participants included in the analysis were mainly female (56%) and Caucasian (88%), with an average age of 51.9 (±13.1) years and a BMI of 24.9 (±3.7) kg/m^2^. The mean follow-up duration was 5.9 (±1.3) years, and 17.2% developed a metabolic syndrome during this time. The OR for developing metabolic syndrome when having moderate-to-severe OSAS was 2.245 (95%CI: 1.214 - 4.149), *p* = 0.010) after adjustments for cohort, age, BMI, and the number of metabolic syndrome components present at baseline. Of note, to assess whether the relationship between OSAS and metabolic syndrome could be bidirectional, a subset analysis was performed on 547 participants free of OSAS at baseline from the Episono cohort. After adjustment for sex, age, and baseline AHI and BMI, metabolic syndrome was not found a significant predictor of incident OSAS.

### 3.3. Mechanism of Action

The immediate consequences of OSAS-related respiratory events are intermittent hypoxia and fragmented sleep. These intermittent hypoxic respiratory events lead to the development of chronic adaptative mechanisms. 

Studies in animals and humans have shown that intermittent exposure to hypoxia leads to sympathetic hyperactivity [64,65], to systemic and vascular inflammation via NFkB [66,67,68,69], to oxidative stress [70], and to pro-inflammatory stimulation of the adipose tissue via hypoxia and hypoxia inducible factor 1 (HIF-1) [71,72]. Sleep fragmentation also disrupts the nychthemeral cycle of cortisol secretion and the somatotropic axis, leading to an increase in nocturnal cortisol and a decrease in IGF-1 [73,74,75,76].

These mechanisms are involved in the development of insulin resistance, endothelial dysfunction, and vascular remodeling that is characterized by increased arterial rigidity. OSAS also alters the nictemeral cycle of arterial pressure: Patients lose the normal nocturnal reduction in arterial pressure of 10%–20% in comparison to daytime pressure, and this is responsible for the so-called ‘non-dipper’ or ‘reverse dipper’ blood pressure profile [49,50,51] which can evolve toward permanent, 24 h hypertension [54,55,56].

The above mechanisms may, at least in part, explain the development of cardiometabolic abnormalities in patients with OSAS. Nevertheless, the relationship between the two disorders is bidirectional (Figure 2): Visceral adiposity is strongly associated with development of OSAS. Central obesity, as described above, is associated with an increase in neck fat infiltration that reduces upper airway volume. Excess intraabdominal fat accumulation increases abdominal pressure, leading to lung volume reduction [5]. In addition to these mechanical effects, visceral adiposity generates low-grade inflammation [77]. Visceral fat adipocytes secrete low levels of TNF-alpha, which then stimulates preadipocytes and surrounding endothelial cells to produce monocyte chemoattract protein-1 (MCP-1). MCP-1, in turn, promotes macrophage recruitment and adhesion to endothelial cells [78] where they secrete proinflammatory cytokines like TNF-alpha, IL-6, and IL-1b, which results in increased plasma C-reactive protein (CRP).

It has been shown that anti-inflammatory therapy can modestly reduce OSAS severity in the absence of other treatments: A placebo-controlled, double-blind study of eight men with obesity and severe OSAS found that a three-week trial of the TNF-alpha antagonist etanercept significantly reduced AHI, IL-6 levels, and objectively measured daytime sleepiness [79]. These data therefore suggest systemic inflammation plays a role in the pathogenesis of OSAS.

## 4. Sleep as a Component of a Comprehensive Lifestyle Intervention

### 4.1. First Steps of Chronotherapy as a Lifestyle Intervention

With regard to the timing of food intake, people with an evening chronotype have a later intake of all three main meals. The timing and daily distribution of this energy intake has been linked to a deleterious cardiometabolic risk profile, as described above. Thus, emerging evidence suggests that meal timing (when) is a dimension of dietary intake that matters, in addition to meal composition (what) and eating behaviors (how) [31,80]. 

Few works have studied the role of food intake timing. One study that included 32 young, normal weight women in a randomized, crossover protocol compared two different lunch-eating conditions: early lunch eating at 13:00 and late lunch eating at 16:30 [81]. Breakfast, lunch, and dinner composition were standardized. Those who ate lunch later had a lower pre-meal resting energy expenditure and glucose tolerance. Early eating, by contrast, was associated with improved circadian cortisol and temperature profiles.

A second study recruited eight overweight individuals who had an average eating duration (time range between the first and last daily food intake) >14 h [82]. The study consisted in a 16-week pilot intervention where participants were asked to restrict their eating duration to a self-selected window of 10–12 h and then to maintain this pattern during both weekdays and weekends to minimize metabolic jetlag. After 16 weeks, the participants had successfully reduced their eating time range and had lost a mean of 3.3 kg (95% CI: 0.9–5.6). Unexpectedly, they also reported greater sleep satisfaction. These results were maintained one year after the intervention. Thus, it appeared that changing the timing of meals not only may affect body weight, resting energy expenditure, and metabolic health but that it may also improve sleep quality.

Fewer studies have investigated the cardiometabolic effect of interventions aiming at improving sleep. A recent review [83] found only seven studies using a variety of interventions to extend sleep that also described the effects of this sleep extension on at least one cardiometabolic risk factor. The research was conducted in both subjects who were healthy and those who were hypertensive. Most had short sleep, although in one study, the subjects had normal sleep (*n* = 14). The interventions were short-term, lasting from 3 days to 6 weeks. These interventions were successful to increase sleep duration from 21 to 177 min. In intervention arms, subjects reported a reduction in their overall appetite, a decreased desire for sweet and salty foods, a lowered daily intake of free sugar, and less caloric intake from protein. Metabolically, insulin sensitivity improved in two studies [84,85].

An additional cross-over study, published since that review, included 21 nondiabetic subjects that usually slept <6 h per night. In the intention-to-treat analysis, the sleep extension group had their sleep extended by 36.0 (45.2) min. There was no improvement in plasma glucose/insulin homeostasis (as measured by oral glucose tolerance tests), but per-protocol analysis of the eight subjects who achieved a sleep duration >6 h during the sleep extension phase showed that sleep extension was associated with improved insulin sensitivity and insulin secretion [86].

Finally, physical activity has also been proven to have a positive impact on both sleep quality and sleep latency, the time in bed before falling asleep, without changing sleep duration [87]. It appears that the timing of physical activity is also important with regard to the cardiometabolic benefit: For instance, postprandial glucose increases were lower if patients with type 2 diabetes carried out physical activity after meals rather than before [88,89]. Thus, the introduction of chronotherapy in either diet or physical activity interventions could improve the cardiometabolic benefits. These interventions do not only change the targeted behavior but also have a positive impact on other behaviors: Changing food timing and increasing physical activity improve sleep quality and promoting sleep extension positively impacts food choices.

### 4.2. Effect of OSAS Treatment

Continuous positive airway pressure (CPAP) is currently the main treatment for moderate-to-severe OSAS. CPAP provides a pneumatic splint to prevent pharyngeal collapse, and it must be applied to the upper airways via a nasal or oronasal mask for ≥4 h per night in order to achieve its therapeutic goals [90]. The beneficial effects of CPAP on daily sleepiness and quality of life have been clearly demonstrated in patients with moderate-to-severe forms of OSAS [91]. 

Whereas the severity of OSAS is clearly parallel to the severity of excess weight [92], CPAP treatment in itself does not allow body weight reduction and was even associated with a small increase in body weight in a meta-analysis of randomized controlled trials (RCTs) [93]. Accordingly, CPAP did not impact visceral adiposity accumulation or body composition in a three-month RCT [94].

Regarding the cardiometabolic impact of CPAP treatment, the most robust evidence is related to blood pressure. Regular CPAP therapy results in modest reductions in blood pressure [95]. This is supported by a recent meta-analysis that pooled the results of 24 h recordings from five randomized trials and found a reduction in systolic and diastolic blood pressure of 4.78 mmHg (IC95%: −7.95 to −1.61) and 2.95 mmHg (IC 95%: −5.37 to −0.53), respectively, in patients with resistant hypertension who had been treated with CPAP [96]. CPAP also modestly improves insulin sensitivity. This was shown in a meta-analysis that included 12 observational studies: CPAP significantly improved the HOMA-IR index [97]. A modest, but significant, effect on insulin sensitivity was also found in a meta-analysis that included 244 patients without diabetes in five RCTs that compared the effects of CPAP and a placebo treatment applied between 6 weeks and 6 months [94,98,99,100]. In patients with type 2 diabetes, CPAP did not improve HbA1c in RCTs, although there was some evidence to suggest that CPAP decreased nocturnal glucose levels and insulin sensitivity [61].

The hypothesis that CPAP might reduce the rate of cardiovascular events was tested in four RCTs. The results all showed that CPAP did not impact the rate of cardiovascular events. This finding was further confirmed by a meta-analysis of 10 RCTs which involved comparison of CPAP to placebo treatments in studies with various main objectives, not only those related to cardiovascular events [101].

Despite the neutral or slightly negative effect of CPAP on body weight per se, it has been suggested that CPAP treatment could be a useful adjunct to improve the overall success of a lifestyle intervention. For example, a diet and physical activity intervention aimed at viscerally obese men (an ancillary study of the SYNERGY trial) found that those who did not have sleep apnea had all-round better results following the intervention than men who had presented with untreated apnea. Despite a similar level of adherence to the diet and physical activity recommendations, the men with apnea showed smaller reductions in BMI, waist circumference, and plasma triglycerides as well as smaller increases in HDL cholesterol and adiponectin [102].

Chirinos et al. [103] have treated people having OSAS by CPAP, weight loss intervention or both combined. When both interventions were cumulated, an incremental reduction in insulin resistance and serum triglyceride levels was obtained compared to weight loss intervention alone. In addition, providing a good adherence to a regimen of weight loss and CPAP resulted in incremental reductions in blood pressure as compared to either intervention alone. 

## 5. Conclusions

This review provides evidence that sleep matters in terms of cardiometabolic health, and more specifically, metabolic syndrome. Sleep deprivation, behaviors linked to an evening chronotype, and social jetlag impact food timing, food preference, and food behavior. In addition, when an individual does not respect their own intrinsic circadian rhythm of rest and activity, dysregulation of hormonal and metabolic regulations occurs, and this strongly contributes to the development of metabolic syndrome.

In addition to sleep habits, the metabolic consequences of OSAS also contribute to metabolic syndrome: Both conditions are linked by a bidirectional autoaggravating relationship through the excess of visceral fat. Central obesity promotes sleep apnea through mechanical action and low-grade inflammation, while OSAS promotes metabolic syndrome through sympathetic nervous overactivity, reactive oxygen production, low-grade inflammation, and alterations in cortisol and IGF-1 circadian fluctuations. 

From these observations, sleep has emerged as a relevant lifestyle factor that needs to be addressed along with diet and physical activity as part of a holistic treatment plan for those with metabolic syndrome. Recent intervention studies have demonstrated that improving one behavior, be it diet, physical activity or sleep, may positively impact the others. Early research on the first interventions targeting sleep extension or food timing seem promising. Further interventional studies should address comprehensive interventions, including chronotherapy through sleep, food, and physical activity timing. Personalized behavioral interventions should be tested in subjects with metabolic syndrome, based on their specific diet and physical activity habits, but also according to their circadian preference.

## Figures and Tables

**Figure 1 nutrients-11-02628-f001:**
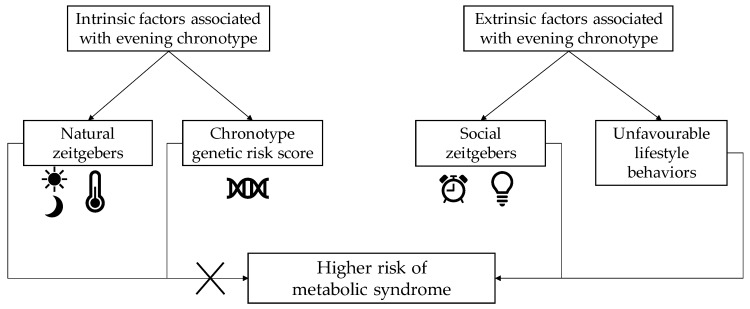
Schema showing the roles of intrinsic and extrinsic factors associated with evening chronotype in the risk of developing metabolic syndrome.

**Figure 2 nutrients-11-02628-f002:**
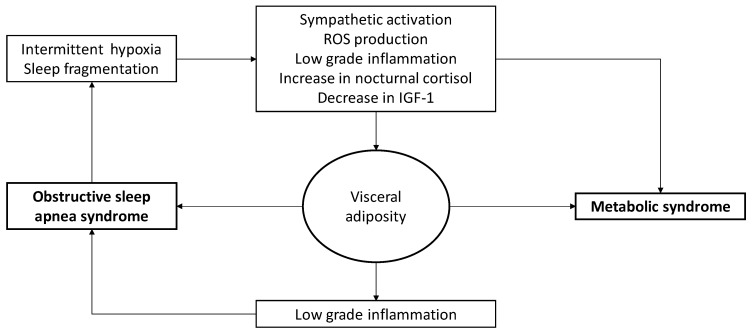
Bidirectional relationship between obstructive sleep apnea syndrome and metabolic syndrome. ROS, reactive oxygen species; IGF-1, insulin-like growth factor 1.

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
