# Peer review of "Sleep Apnea and Sleep Habits: Relationships with Metabolic Syndrome"

_nutrients, 2019, doi:10.3390/nu11112628_

Round 1
Reviewer 1 Report
-line 18: OSAS has to be explained in the abstract before using the abbreviation.
-line 38: "clearance" ?
-line 63: MSF, abbreviation, first explain
-line 134 more the "M" should be printed in small
-line 161: why using the German word "zeitbrekers", also in the Figure 1.
-line 165: this, capital T
-line 215: space between "therisk"
-lines 247 up to 263 use the same spaciation as all over the article
-line 283: "metabolic" small "m"
-omit the lines 432-434.
As a general remark this article has to be submitted to another type of journal. The link and relation to food or nutrients is that thin.
Author Response
Response to Reviewer 1 Comments
I thank the reviewer for his/her time and efforts in reviewing this work.
-line 18: OSAS has to be explained in the abstract before using the abbreviation.
Thank you, it has been done.
-line 38: "clearance" ?
Mistake corrected by “metabolize”
-line 63: MSF, abbreviation, first explain
Corrected
-line 134 more the "M" should be printed in small
Corrected
-line 161: why using the German word "zeitbrekers", also in the Figure 1.
In the field, the German word “zeitgeber” or “timecue” is accepted from the earliest studies that were performed in the field by German scientifics. The German term has been put between “..”.
-line 165: this, capital T
Corrected
-line 215: space between "therisk"
Corrected
-lines 247 up to 263 use the same spaciation as all over the article
Corrected
-line 283: "metabolic" small "m"
Corrected
-omit the lines 432-434.
Done except that I add an acknowledgment for English revision.
As a general remark this article has to be submitted to another type of journal. The link and relation to food or nutrients is that thin.
I was an invited paper for a special issue.

Reviewer 2 Report
This appears to be a non-systematic review of the literature regarding associations between sleep apnea, other sleep dimensions, and metabolic syndrome. While it has the potential to be an important contribution to the literature, this review, as described in its current form is not thoroughly detailed enough to be replicable.
Suggestions for the author before any additional evaluation of the results are as follows:
Please provide a methods section that contains important study design information including the following: population under study, exposure, comparison considered, definition of metabolic syndrome, time period of published studies that were reviewed, indicate the review type (e.g., non-systematic, narrative), inclusion and exclusion criteria for the studies included in this review, and the process of selecting articles.
Please provide citations for the definitions presented. Please define metabolic syndrome in the context of this review.
Sleep debt usually includes Friday and Saturday nights as weekend sleep duration because Sunday night is actually considered a work week night.
Please review page 2, lines 58-59.
There were no acknowledgements on page 11, lines 432-434.
Author Response
Response to Reviewer 2 Comments
I thank the reviewer for his/her time and efforts in reviewing this work.
This appears to be a non-systematic review of the literature regarding associations between sleep apnea, other sleep dimensions, and metabolic syndrome. While it has the potential to be an important contribution to the literature, this review, as described in its current form is not thoroughly detailed enough to be replicable.
Suggestions for the author before any additional evaluation of the results are as follows:
Please provide a methods section that contains important study design information including the following: population under study, exposure, comparison considered, definition of metabolic syndrome, time period of published studies that were reviewed, indicate the review type (e.g., non-systematic, narrative), inclusion and exclusion criteria for the studies included in this review, and the process of selecting articles.
The reviewer is right that the present work is not structured as a systematic review and thus has currently no “method” section. It was and invited paper and guest editors for the special issue requested a narrative review.
I have added in the end of the “introduction” paragraph that the work is a narrative review that does not require a method section.
Please provide citations for the definitions presented. Please define metabolic syndrome in the context of this review.
Citations have been added were needed. A paragraph regarding metabolic syndrome definition has been added at the beginning of paragraph 2.
“Metabolic syndrome: In 1998, the World Health Organization (WHO) became the first organization to introduce the term metabolic syndrome, with a primary focus on insulin resistance and hyperglycemia [3]. In 2001, the National Cholesterol Education Program’s Adult Treatment Panel III (NCEP-ATP III) released its own definition, adding abdominal adiposity, specifically an increased waist circumference, as a major component of the syndrome [4]. Several definitions followed, issued from different societies, that mainly diverged on the clinical evaluation of abdominal adiposity. In 2009, the International Diabetes Federation Task Force on Epidemiology and Prevention; the National Heart, Lung, and Blood Institute; the American Heart Association; the World Heart Federation; and the International Atherosclerosis Society joined to release a statement harmonizing the criteria for defining the metabolic syndrome. This is the definition that is in use today, and it takes into account population-specific cutoffs for waist circumference [5]. Metabolic syndrome is defined by 3 of 5 criteria with dyslipidemia being 2 criteria among: Fasting glucose ≥100 mg/dL or anti-diabetic therapy, increased waist circumference, TG ≥150 mg/dl and/or HDL-C <40/50 in men/women or anti-lipidic therapy, ≥130/85 mmHg or therapy.”
Sleep debt usually includes Friday and Saturday nights as weekend sleep duration because Sunday night is actually considered a work week night. Please review page 2, lines 58-59.
Indeed, my mistake. Corrected.
There were no acknowledgements on page 11, lines 432-434.
Corrected

Round 2
Reviewer 1 Report
They have followed up my suggestions.
Reviewer 2 Report
Even though this is a narrative review, details about the process of completing the review are still needed. The review remains non-replicable. The authors did not adequately address my concerns.